# Intergenerational Transaction of Emotional Health in Collective Family Context: Family Functioning, Parental and Children’s Gratitude, and Their Depression

**DOI:** 10.3390/healthcare13020147

**Published:** 2025-01-14

**Authors:** Jerf W. K. Yeung

**Affiliations:** 1Department of Social and Behavioural Sciences, City University of Hong Kong, Hong Kong, China; ssjerf@gmail.com; 2Graduate School of Human Sciences, Osaka University, Osaka 565-0871, Japan

**Keywords:** family functioning, gratitude, depression, parent–child dyads, transactional model

## Abstract

**Background:** The current study is intended to examine how positive family functioning, collectively constructed by parents and children in the family context, may contribute to their gratitude and depression, two pivotal human emotions closely related to human health, in which the gratitude and depression of parents and children are assumed to affect each other bidirectionally and parental and children’s gratitude is expected to link the relations between positive family functioning and their depression. **Methods:** The data for analysis came from a community sample of 310 Chinese parent–child dyads, which were measured by the Family Functioning Style Scale (FFSS), Gratitude Questionnaire (GQ), and depression subscale of the Brief Symptom Inventory (BSI) from both the parent and child participants. **Results**: The results indicated that positive family functioning contributed to higher parental and children’s gratitude and their less depression, and parental and children’s gratitude and depression affected each other positively. Furthermore, serial mediation analyses discovered the complex processes from positive family functioning to the actor’s depression in parents or children through their gratitude or the links from the partner effect to the actor effect of parental and children’s gratitude or from the partner effect of parental or children’s gratitude to their partner effect of depression. **Conclusions:** Findings of the current study prove the collective effects of family functioning on the emotional development of gratitude and depression among parents and children, in which parental and children’s gratitude and depression bidirectionally impact each other and lead to the actor’s depression in them through the complex serial mediated effects. For this, pro-family and health-related policies and services should be provided to promote family functioning and emotional management in a home context to enhance family and emotional health among parents and children.

## 1. Introduction

Human emotional development can be understood within the context of cognitive processes [1], in which the social experiences and interpersonal interactions we encounter are directly and profoundly related to our formulation of cognitive development that leads to emotional health. Undeniably, family relationships, interactions, and communication that are commonly referred to as family functioning may construct the most intimate and fundamental socialization and life experiences to shape the cognitive processes of family members [2,3], contributing to their emotional outcomes. For this, it is pivotal to investigate how family functioning collectively constructed by different family members, e.g., parents and children, may shape their emotional development through cognitive processes. Bell et al. [4] mentioned that “any conceptualization of emotional development is incomplete without accompanying consideration of cognition (p. 376).” For this, the current study is intended to investigate the emotional development of gratitude and depression among parents and children in the context of family functioning because these two emotional outcomes have close relevance to human mental and physical health [5,6,7]. This is important as family functioning collectively formed and fostered by family members with concerted efforts in a home context may impact their cognitive processes and emotional development concomitantly due to the similar relational, interactive, and communication patterns they commonly share [2,8]. To the author’s knowledge, no study has investigated the aforementioned study relations. In fact, existing pertinent research mainly employed a unidirectional and parent-driven approach to study the emotional development of gratitude and depression in children and only examined how the emotional health of parents shaped their parenting practices or parental behaviors to affect the emotional outcomes in their children [9,10]. For example, researchers tended to study the adverse impacts of parental depression on children’s development of depression through parenting behavior and/or parent–child relationships [3,10], which have omitted how the collective effects of family context may impact both parental and children’s emotional health and also skipped the cross-domain intergenerational transaction of emotional health in each other between parents and children.

As informed by the model of cognition–emotion integration and recent family research findings [8,11,12], both cognition and emotional development of family members are first cultivated and formed in the relational and interaction processes and experiences within the family realm and then sway their performances of the cognitive and emotional responses in external social settings. It is expected in this study that family functioning collectively constructed by parents and children may lead to their development of gratitude and depression. Specifically, the current study aims to investigate how positive family functioning aggregately constructed by parents and children, which refers to the manifestation of supportive family relationships, caring interactions, and efficient communication among family members in the family realm, may contribute to both parental and children’s gratitude and depression positively and negatively. Due to the incompatible nature of gratitude and depression in the perspective of Beck et al.’s cognitive depressive bias and the empirical predictive power of gratitude to depression [13,14,15], this study is also planned to examine whether parental and children’s gratitude would negatively predict their depression, in which both the emotional outcomes are expected to be affected by positive family functioning. Moreover, the transactional model of cognition and emotional development posits that family members may mutually and bidirectionally affect their psychological and behavioral performances [16]. For this, the current study is intended to test whether parental gratitude and depression would positively lead to the development of children’s gratitude and depression and vice versa. Lastly but importantly, based on the structural relations between family functioning, parental and children’s gratitude, and their depression considered above, this study is also intended to test the serial mediations of the actor and partner effects of parental and children’s gratitude and the partner effects of parental and children’s depression on the actor’s depression in parents or children for dissecting the complex relations between the transactional development of parental and children’s gratitude and depression formulated in the family context.

## 2. Theoretical Framework of the Study

Family is the most important nurturing and protective socialization and formative agent related to the mental and physical health of its members. Empirical research has reported that healthy and supportive family relationships, interactions, and communication among family members in a home context are promotive of not only their life satisfaction [17], psychological well-being [18], and happiness [19] but also can alleviate depressive symptoms [20], anxiety [21], and emotional distress [22]. As aforementioned, this corresponds with the transactional model stressing that the psychological and behavioral performances of family members are mutually reinforced and influence each other [16], which, in the current study, suggests that family functioning is collectively constructed and established among parents and children in the family realm is expected to affect their emotional development, e.g., gratitude and depression. Apparently, individual interpersonal relationships and communication experiences can act as a cognitive impetus to cultivate grateful thoughts and notions [23,24], which means that if a family environment flourishes with supportive relationships, constructive interactions, and efficient communication among family members, a manifestation of positive family functioning, family members there would have a higher sense of gratitude [25,26] and fewer depressive symptoms [14,27]. The development of gratitude is described by Gordon, et al. [28] as a form of interpersonal emotion regulation being nurtured and fostered in the context of supportive, sharing, and mutually caring interpersonal relationships and experiences, especially in the family context. This is consonant with cognitive appraisal theory, positing that environmental conditions, e.g., positive family functioning, can activate our retrospective cognitive processes in the enhancement of the acknowledgment of what we have and encounter in an appreciative and thankful way [29], which are important to the development of gratitude. Although Rodrigues et al. [30] have mentioned that “family relationships have been proposed as an important context for the development of gratitude (p. 263)”, existing research has rarely examined how family functioning contributes to the development of parental and children’s gratitude concomitantly. In fact, limited studies have investigated how parent–child interactions shaped the development of children’s gratitude, leaving the relations between family functioning and the development of gratitude in both parents and children uncharted. As gratitude is related to interpersonal emotion regulation, which means that gratitude can be cultivated and nurtured in a supportive and constructive relational and interpersonal context within the family realm, the current study anticipated that positive family functioning would promote both parental and children’s gratitude.

In addition, the development of depression can also be described in the cognitive processes embedded in interpersonal and social contexts [7], e.g., family functioning. Although the occurrence of depression is empirically reported as relating to genetic and neurobiological inheritance [31,32], this explication does not largely account for the etiology of the risk [33], leaving the unexplained variances reasonably ascribable to the external social contexts, such as interpersonal relationships and interactions in the family realm. Manifestly, the family context is the most important cognitive contributor to people’s mental representations and internal models related to emotional development [34]. If family members live in a home environment of a supportive, caring, and intimate climate, they will cultivate better cognitive competence to facilitate emotion regulation in prevention of the adverse impacts of negative emotions [20,35], e.g., depression. The cognitive appraisal theory assumes that positive predictability, reliability, and expectation established in interpersonal relationships and interactions in a home context, e.g., positive family functioning, are the pivotal cognitive base to promote individual self-worthiness and problem-solving abilities in regulating emotions competently and adequately [29], which are important to preclude the occurrence and harmful effects of depression. Lam and Chen [14] reckoned that “depressive symptoms (are) a negative interpersonal emotional experience that is associated with individuals’ social relationships and particularly derives from the family ambiance (p. 1310).” Relevant research reported that healthy parent–child relationships, interactions, and attachment are directly related to fewer depressive symptoms in children. In Lam and Chen’s study [14], they found that constructive family interaction in terms of efficient communication, mutuality, and harmony was significantly predictive of fewer depressive symptoms among Chinese adolescent children. In addition, Keles, et al. [36] found that family support and harmonious marital relationships significantly predicted a lower level of depression among perinatal parents. Furthermore, Gomez-Baya et al. [20] reported that the declining quality of father–child and mother–child relationships were significantly related to an increase in depressive symptoms in a sample of Spanish youths during adolescence. Nevertheless, to the author’s knowledge, no study has concomitantly investigated how family functioning collectively constructed by parents and children may contribute to their development of depression. Accordingly, the current study anticipated that positive family functioning would predict less depression among both parents and children.

In emotional development research, individual gratitude was consistently found to predict fewer depressive symptoms [37,38,39,40]. Putting this relation in the model of family socialization of emotional regulation can help elucidate the explanatory mechanism of why gratitude is negatively related to depressive symptoms [41]. Specifically, positive family functioning is able to help create an emotional climate in the family that can beneficially facilitate the cognitive process for building a healthy mental representation and internal model in family members conducive to their development of grateful thoughts and alleviation of the occurrence and harm of negative emotions [4], e.g., depression. In fact, people who are more grateful tend to be less depressed due to their higher life satisfaction, contentment, active coping, and resilience [39,42,43]. Evidently, Lam and Chen [14] found that gratitude was not only significantly and directly predictive of a lower level of depression among young adults but also mediated the relationship between constructive family interaction and the depressive outcome in these young adults. Recently, Rodrigues et al. [30] reported that secure attachment relationships with mothers and fathers were significantly and negatively associated with fewer depressive symptoms in adolescent children through the mediation of dispositional gratitude. In their recent meta-analysis, Iodice et al. [44] reported that the effect size between gratitude and depression was *r* = −0.39, indicating that people of greater gratitude tend to have fewer depressive symptoms in a substantial magnitude. Taken together, the current study anticipated that positive family functioning would predict a higher level of gratitude and a lower level of depression among parents and children, in which parental and children’s gratitude would link the relations between positive family functioning and their depression, meaning that parental and children’s gratitude would predict less depression.

When examining the relation of gratitude and depression in the family context, existing research tends to mainly take a unidirectional and parent-driven approach in analyzing the influences of parental gratitude and/or depression on their children’s development of these emotional outcomes [14,30]. Recent, more advanced longitudinal studies reported that the emotional development of parents and children in the family realm is mutually reinforced and impacted by each other [45,46,47]. Although the intergenerational transmission model is supported by empirical findings, in which the transfer of individual dispositions, traits, behaviors, and emotional outcomes from parents to their children is reported to be sizable [10,48], children’s emotional and behavioral responses in the family context are also found to sway their parents’ emotional and behavioral development [47,49,50]. Correspondingly, the theory of emotion coregulation and the transactional model suggest that the emotional patterns between parents and children are bidirectional through the cognitive processes of mutual attunement and reinforcement subconsciously rather than unidirectional from parents to children [51,52]. However, researchers have only examined how parental gratitude and/or depression lead to the development of children’s gratitude and/or depression, omitting the importance of their bidirectionality and mutuality between parents and children. For example, as mentioned above, Lam and Chen [14] reported that family interaction significantly predicted children’s gratitude and depression, in which children’s gratitude mediated the relationship between family interaction and their depression. Similarly, Rodrigues et al. [30] reported in their study that secure attachment relationships with parents significantly contributed to children’s dispositional gratitude and depressive symptoms, and dispositional gratitude acted as a mediator linking with the association between parental attachment and children’s depression. Due to a paucity of research having analyzed how parental and children’s gratitude and depression may impact on each other bidirectionally in the family context by a single study, the current study is also aimed to investigate this relation by anticipating that positive family functioning would predict higher parental and children’s gratitude and their less depression, in which parental gratitude and depression would positively contribute to children’s gratitude and depression, and vice versa.

## 3. The Current Study

In all, the current study considers family functioning as a collective socialization and formative agent that impacts emotional development in terms of gratitude and depression among both parents and children. Specifically, this study anticipated that positive family functioning would directly contribute to higher gratitude and less depression among parents and children. In addition, due to the incompatible nature of gratitude and depression suggested by the thesis of Beck et al.’s cognitive depressive bias and the preventive effects of gratitude on depression reported [13,39,53], this study tests whether parental and children’s gratitude would link the relation between positive family functioning and their depression, in which parental and children’s gratitude are expected to negatively predict their depression. Furthermore, considering the perspectives of emotion coregulation and the transactional model [16,51], the current study also examines whether parental gratitude and depression would positively predict their children’s gratitude and depression, respectively, and vice versa. According to the above-hypothesized structural relations between positive family functioning, parental and children’s gratitude, and their depression, it is of research importance to test whether parental and children’s gratitude solely or through the link from the partner effect to the actor effect of parental and children’s gratitude or through the link from the partner effect of parental or children’s gratitude to their partner effect of depression mediates the relation between positive family functioning and the actor’s depression in parents or children. For example, the path from parental gratitude (partner effect) to children’s gratitude (actor effect) or the path from parental gratitude (partner effect) to parental depression (partner effect) would mediate the relation between positive family functioning and children’s depression (actor’s outcome), and vice versa when testing parental depression as an actor’s outcome. In sum, this study has the following hypotheses:

**H1.** 
*Positive family functioning would predict higher gratitude and lower depression among parents and children.*


**H2.** 
*Gratitude and depression of parents would positively predict their children’s gratitude and depression, respectively, and vice versa.*


**H3.** 
*Gratitude of parents and children would negatively predict their own depression.*


**H4.** 
*The actor effect of parental and children’s gratitude would mediate the relation between positive family functioning and their own depression.*


**H5.** 
*The partner effect of parental and children’s gratitude would work with the actor effect of parental and children’s gratitude or the partner effect of parental and children’s depression to mediate the relation between positive family functioning and the actor’s own depression in parents or children.*


In modeling and analyzing the above-hypothesized study relations, pertinent sociodemographic covariates of parents and children are adjusted, which include gender, age, and educational attainment of parents and children. This is because mothers and female children tend to be more grateful but also exhibit more emotional distress [7,34,54,55]. In addition, older-aged parents and children are reported to have greater gratitude and fewer depressive symptoms [6,7,37,56]. Moreover, people of higher educational levels generally have greater gratitude and lower depressive symptoms [7,34,57].

## 4. Methods

### 4.1. Sample and Data

The data for analysis in the current study was based on a community sample of 310 Chinese parent–child dyads who were recruited with the help of the Hong Kong Young Women’s Christian Association (HKYWCA), one of the largest and long-history NGOs in Hong Kong. The parent participants were the main caregivers in the family, ensuring their close interactions and influences on the development of their children. The selection criteria for sampling the parent–child dyads include that the child participants in the participating families should be in the age range between 13 and 21 years old, which means they were in their critical development period, susceptible to the influences of family context, and the parent participants should be the main caregivers to ensure their knowledge of the family context and the development of their children. In addition, if the participating family had two or more target adolescent children eligible for the study, the one who had just passed her or his birthday should be selected, which was to increase the variance of the child participants. However, if the family had twin eligible children suitable for the study, the firstborn should be selected to increase randomization. In the sampling procedures, the purposive sampling approach was adopted to align with the sampling selection criteria mentioned above, in which the author first introduced the management of HKYWCA to the study purpose and sampling selection criteria set for this study. After that, directors and social workers in respective service units of HKYWCA helped to invite eligible families to take part in the study. These service units cover family services, school social work, and youth service centers in HKYWCA.

During data collection, the parental and child versions of the printed questionnaires were enclosed in a survey package and sent to the participating families that were eligible for the study through HKYWCA. In the data collection process, social workers in HKYWCA first helped publicize and recruit eligible Chinese parent–youth dyads who were willing to participate in the study. Then, the parent version and child version questionnaires coded with the same identifiers were sent to the participating families, respectively. After completion of the questionnaires, each participating family was rewarded with a cash purchase coupon worth HKD 100, compensating them for their efforts and time used in participating in the study. After data collection, the management of HKYWCA arranged to send back the completed parent- and child-version questionnaires to the author of the current study. The study was approved by the ethical review committee of City University of Hong Kong.

### 4.2. Measures

**Positive family functioning** was measured by the 26-item Family Functioning Style Scale (FFSS) [8], which has been commonly used to tap into constructive relationships, healthy interactions, efficient communication, and mutual support among family members in a home environment of Chinese societies with good measurement validation [8,58]. Example items include “We take pride in even the smallest accomplishments of family members” and “Our family sticks together no matter how difficult things get.” To enhance the objectivity and accuracy of evaluating positive family functioning that was collectively established by parents and children in the family context, a multi-informant approach was adopted to collect data from both the parent and child participants. This measurement approach is designed to collect data from related parties who may have different perceptions regarding the same measurement concept(s) being assessed [59], which means both the responses from the parent and child participants were collected and averaged to construct a combined measure of positive family functioning. This approach has been verified to increase external validity and reduce the bias of common method variance [59,60,61]. In this study, excellent Cronbach alphas were obtained for the parent–child dyads, which were α = 0.963 and 0.964 for the parent and child participants. The McDonald’s omega coefficients were also reflective of excellent measurement reliability for both the parent and child participants, ꞷ = 0.961 and 0.964.

**Gratitude of parents and children** was measured by the 6-item Gratitude Questionnaire (GQ) [62], which is a well-validated measure and has been used in Chinese societies to measure grateful thoughts and mentality [63,64]. The reliability values for the parent and child participants were α = 0.821 and 0.837, respectively, which indicate an adequate level. The McDonald’s omega coefficients for the parent and child participants were also highly reliable, ꞷ = 0.853 and 0.835.

**Depression of parents and children** was measured by the 6-item depression subscale of the Brief Symptom Inventory (BSI) [65], which has been commonly used to measure psychological distress and mental disorders among various populations [66,67,68] and has been well validated in the Chinese context [69]. The Cronbach alpha coefficients were excellent in this study for both the parent and child participants, which were α = 0.932 and 0.885, respectively. The McDonald’s omega reliability coefficients for both the parent and child participants were ꞷ = 0.931 and 0.893, respectively, indicating an excellent level of internal consistency.

**Sociodemographic covariates of parents and children** include their gender, age, and educational attainment. Gender of parents and children is a dichotomous variable (1 = male, 2 = female), and the age of parents and children is a count variable. In addition, educational attainment of parents and children is an ordered categorical variable, in which the educational variable for parents are coded: 1 = primary school or below, 2 = secondary school, 3 = associate degree or diploma, 4 = university degree, and 5 = postgraduate degree; and the educational variable for children is coded: 1 = primary school or below, 2 = secondary one, 3 = secondary two, 4 = secondary three, 5 = secondary four, 6 = secondary five, 7 = secondary six, and 8 = postsecondary school.

### 4.3. Analytic Techniques

In this study, structural equation modeling in combination with serial mediation analysis procedures was used to investigate how positive family functioning contributes to parental and children’s gratitude and depression and the bidirectional impacts of these emotional outcomes among parents and children on each other, as well as their mediated effects. Specifically, positive family functioning was treated as an exogenous variable to predict both parental and children’s gratitude and depression that were considered endogenous variables, in which parental and children’s gratitude was modeled to link the relations between positive family functioning and their own depression. In addition, serial mediation analysis was used to test whether parental and children’s gratitude mediate the relation between positive family functioning and their own depression and whether the partner effect of parental or children’s gratitude is through the actor effect of parental or children’s gratitude or through the partner effect of parental or children’s depression on the actor’s depression in parents and children. The modeling procedures were fit in M*plus* 8.7 by maximum likelihood estimation with Monte Carlo integration by setting <Integration = Montecarlo> [70]. This estimation approach is more effective for evaluating the chain mediation effects in interrelated structural relations [71,72]. Model fit was evaluated by comparative fit index (CFI), root mean square error of approximation (RMSEA), and standardized root mean square residual (SRMR), in which an acceptable model fit is CFI > 0.90, RMSEA < 0.08, and SRMR < 0.1, and an excellent model fit is CFI > 0.95, RMSEA < 0.06, and SRMR < 0.08 [73]. The M*plus* code for modeling is provided in Table A1 in Appendix A.

## 5. Results

Table 1 describes the sociodemographic characteristics of the study sample of 310 parent–child dyads, in which main caregiver mothers were the main parent participants that shared 79.7% (n = 247), and main caregiver fathers had 20.3% (n = 63). The mean age of parents was 46.280, meaning that they generally were in middle age. In addition, the average educational attainment of parents was 2.374, indicating that they were generally of secondary school education. For the child participants, female adolescent participants (58.7%, n = 182) were a little more than their male adolescent counterparts (41.3%, n = 128), and their average age was 16.022, meaning that they were in their mid-adolescence. Lastly, the educational level of child participants was 4.358, denoting that they were generally at secondary school three level. In addition, Table 2 presents the partial correlations of the study variables controlling for the gender, age, and education of parents and children. Specifically, positive family functioning significantly correlated with higher parental and children’s gratitude, *r* = 0.298 and 0.256, *p* < 0.001, and also correlated with less parental and children’s depression, *r* = −0.390 and −0.244, *p* < 0.001. Moreover, parental gratitude was significantly and positively correlated with their children’s gratitude, *r* = 0.231, *p* < 0.001, and negatively correlated with their own depression and children’s depression, *r* = −0.250 and −0.127, *p* < 0.001 and 0.05. Moreover, children’s gratitude was significantly and negatively correlated with their parents and their own depression, *r* = −0.154 and −0.382, *p* < 0.01 and 0.001, in which parental depression was significantly and positively correlated with their children’s depression, *r* = 0.225, *p* < 0.001.

The structural relations of positive family functioning in prediction of children’s depression through parental and children’s gratitude and parental depression were first analyzed (Model 1), in which an excellent model fit was obtained: CFI = 0.989, RMSEA = 0.025, SRMR = 0.034. Figure 1 shows the standardized effects, in which positive family functioning was significantly predictive of higher parental and children’s gratitude, *β* = 0.329 and 0.223, *p* < 0.001, and also fewer parental and children’s depression, *β* = −0.343 and −0.121, *p* < 0.001 and 0.05. In addition, parental gratitude and depression significantly and positively contributed to their children’s gratitude and depression, *β* = 0.178 and 0.130, *p* < 0.01 and 0.05, respectively. Moreover, the gratitude of parents and children was significantly and negatively predictive of their own depression, *β* = −0.145 and −0.361, *p* < 0.05 and 0.001. This model explained 21.5% and 21.2% of the variances of children’s and parental depression and 17.4% and 11.9% of the variances of children’s and parental gratitude. Table 3 shows the results of serial mediational analysis, in which parental and children’s gratitude significantly mediated the relations between positive family functioning and their own depression, *β_ind_* = −0.043 and −0.079, *p* < 0.05 and 0.001. Moreover, the relation between positive family functioning and children’s depression was significantly mediated by parental depression, *β_ind_* = −0.047, *p* < 0.05, and the path from parental gratitude to children’s gratitude, *β_ind_* = −0.020, *p* < 0.01. Moreover, the path from parental gratitude to parental depression also marginally significantly mediated the relation between positive family functioning and children’s depression, *β_ind_* = −0.006, *p* < 0.1.

In addition, Model 2 analyzed the structural relations of positive family functioning in prediction of parental depression through children’s and parental gratitude and children’s depression, in which an excellent model fit was also obtained: CFI = 0.991, RMSEA = 0.021, and SRMR = 0.034. Results in Figure 2 show the standardized effects that positive family function significantly predicted higher children’s and parental gratitude, *β* = 0.282 and 0.287, *p* < 0.001, and less children’s and parental depression, *β* = −0.171 and −0.310, *p* < 0.01 and 0.001. Moreover, children’s gratitude and depression significantly and positively predicted parental gratitude and depression, *β* = 0.166 and 0.138, *p* < 0.01 and 0.05, respectively. Furthermore, both children’s and parental gratitude significantly contributed to their lower depression levels, *β* = −0.368 and −0.135, *p* < 0.001 and 0.05. This model explained 22.8% and 20.5% of the variances of parental and children’s depression and 14.5% and 14.3% of the variances of parental and children’s gratitude. Serial mediation analysis found that children’s and parental gratitude significantly mediated the relations between positive family functioning and their own depression, *β_ind_* = −0.104 and −0.039, *p* < 0.001 and 0.05. Moreover, the relation between positive family functioning and parental depression was significantly mediated by children’s depression, *β_ind_* = −0.024, *p <* 0.05, and the path from children’s gratitude to children’s depression, *β_ind_* = −0.014, *p* < 0.05. Moreover, the path from children’s gratitude to parental gratitude also marginally significantly mediated the relation between positive family functioning and parental depression, *β_ind_* = −0.006, *p* < 0.1.

For the effects of sociodemographic covariates regarding parental and children’s gender, age, and education on their gratitude and depression in Model 1 (see Table A2 in Appendix A), parental and children’s gender significantly predicted their own gratitude negatively and positively, which indicates that mother caregivers compared to their father caregivers had a lower level of gratitude, *β* = −0.125, *p* < 0.05, and female children compared to their male counterparts, however, had a higher level of gratitude, *β* = 0.136, *p* < 0.01. Nevertheless, the education of both parents and children significantly and positively predicted their gratitude, meaning that both parents and children with a higher level of education demonstrated more gratitude, *β* = 0.171 and 0.223, *p* < 0.01 and 0.05. In Model 2, parental and children’s gender marginally significantly and significantly predicted their own gratitude negatively and positively, which denotes that mother caregivers compared to their father caregivers had a lower level of gratitude, *β* = −0.095, *p* < 0.1, and female children compared to their male counterparts had a higher level of gratitude, *β* = 0.129, *p* < 0.01. Moreover, children’s education in Model 2 significantly predicted their higher gratitude, *β* = 0.219, *p* < 0.05, but parental education did not have a significant effect. Moreover, parental education significantly predicted their own depression positively, *β* = 0.153, *p* < 0.01, meaning that parents of a higher level of education also had more depression.

## 6. Discussion

Human emotional development is substantially susceptible to the impacts of environmental conditions, in which family is the most direct and profound socialization and formative agent influential on various emotional outcomes [41]. This study is the first attempt to investigate how family functioning collectively constructed by parents and children may contribute to their gratitude and depression, in which parental and children’s gratitude and depression are believed to bidirectionally affect each other and mediate the relation between positive family functioning and the actor’s depression in parents and children. Evidently, the results of the current study found that positive family functioning did significantly predict parental and children’s higher gratitude and less depression. In fact, gratitude and depression are important positive and negative emotions [5,7,53] that bear prolonged influences on human physical and psychological health. This is because emotional development and responses are directly related to human neurobiological, behavioral, psychological, and physical representations and health [45,74,75]. Therefore, analyzing and clarifying how the family context, e.g., family functioning, affects gratitude and depression in parents and their children is an important research contribution to the literature [2]. Gratitude is an inherent emotional disposition related to one’s expression of thankfulness and appreciation of what they have and encounter in life [62]. Research has found that gratitude is associated with a number of benefits, such as increased optimism, self-confidence, resilience, happiness, life satisfaction, and physical health [56], and can also preclude potential emotional harms, such as depression, anxiety, psychological distress, and suicidality [56,76]. The current study found that the collective and constructive family context, that is, positive family functioning, is significantly promotive of gratitude among parents and children, which then contributes to their less depression. These findings respond to the importance of cultivating a supportive and healthy interpersonal context in the family realm to help family members establish positive cognitive processes for their development of gratitude and alleviation of depression in the processes of implementing healthcare policies and services. This includes letting family members value the importance of cohesive and supportive family relationships, appreciating the proper ways of life experiences and encounters in the family context, and establishing appropriate strategies to promote their positive emotions and handle negative emotions properly.

Moreover, both parental and children’s gratitude are found to predict their decreased depression levels, which concur with existing research results in support of the counter nature of gratitude and depressive symptoms proposed by the cognitive depressive bias perspective [13]. This is because gratitude is a quality of being thankful and showing appreciation for what one has encountered and experienced, which is related to returning kindness to the actor and people and external environments around her or him [6]. Manifestly, gratitude is deemed a type of positive emotion that can not only broaden immediate thinking to a more transcendent and appreciative way for avoiding the happening of depression [15,53] but also helps employ and utilize personal and social resources flexibly [77], e.g., individual strengths, intellectual and psychological capabilities, and interpersonal relationships. All these can abate the impacts of negative events and failures on incurring depressive symptoms. Moreover, pertinent research found that gratitude is positively related to emotional intelligence, which is believed to help alleviate depression [78,79]. For this, gratitude should be promoted in the family realm as a positive emotion to prevent depression. In fact, the characteristics of depressive symptoms include low mood, inability to concentrate, feeling worthlessness, aversion to activity, excessive guilt, and recurrent thoughts of death [7,37], which may severely impair one’s physical, mental, and interpersonal health and well-being. Therefore, family and healthcare practitioners ought to design and implement programming and services at individual, family, and community levels for the enhancement of positive emotions, including gratitude, to tackle the prevalent harms of depression. This is believed to be an effective strategy before medication and clinical interventions are needed as the therapeutic end for clinical purposes. This is important as the etiology and occurrence of depression result from a complex interaction process related to the social, psychological, and biological spheres [7,14], in which people who have gone through adverse life events are more susceptible to the development and harm of depression.

Furthermore, the current study has attempted to investigate whether parental and children’s gratitude and depression may affect each other, which, to the author’s knowledge, is the first research attempt to scrutinize the bidirectional reinforcement of emotional health of gratitude and depression between parents and children. This research attempt is needed as researchers tended to adopt a unidirectional and parent-driven approach in studying emotional development in the family context [9,10], which means that the development of gratitude and depression in a family is generally regarded as a transmission from parents to children. Although relevant research has supported this unidirectional and parent-driven approach to the emotional development of children in the family context [14,25,30], the transactional model asserts that the emotional and behavioral representations are mutually affected and shaped between parents and children in the family context [16]. Results of the current study support the thesis of the transactional model that parental gratitude and depression contribute to their children’s gratitude and depression, and children’s gratitude and depression also lead to the development of their parental gratitude and depression. Accordingly, although we conventionally consider parents to play the dominating role in shaping the emotional health of their children, the results of the current study support the influences of children’s gratitude and depression on their parents’ emotional health. Thus, policymakers, educators, and family and healthcare practitioners should take note of this complex and bidirectionally reinforcing family process regarding the mutuality between parents and children when designing and implementing family policies and interventions. Of specific importance, family and healthcare practitioners should consider interventions and services at both the parental and child levels concomitantly to promote their constructive and healthy interactions and communication as well as emotional management in the family context, which are pivotal to enhancing family and emotional health concertedly.

Besides, serial mediational analyses showed that parental and children’s gratitude not only significantly mediated the relation between positive family functioning and their own depression but also acted as a partner effect in work with the actor effect of parental or children’s gratitude and the partner effect of parental or children’s depression to mediate the relation between positive family functioning and the actor’s depression in parents or children. For Model 1, in which children’s depression acts as the emotional outcome, both children’s own gratitude and the path from parental gratitude to children’s gratitude significantly mediated the relation between positive family functioning and children’s depression. In addition, the path from parental gratitude to parental depression also marginally significantly mediated the relation between positive family functioning and children’s depression. For Model 2, in which parental depression is treated as the emotional outcome, the paths from children’s gratitude to children’s depression and children’s gratitude to parental gratitude significantly and marginally significantly mediated the relation between positive family functioning and parental depression. Moreover, the partner’s depression in Model 1 and Model 2, which are parental depression in Model 1 and children’s depression in Model 2, significantly mediated the relation between positive family functioning and the actor’s depression. These serial mediational analyses reveal the complex and linked lives of family members in the family context, in which their emotional development is not only affected by family functioning collectively but also interacts with each other to impact the emotional development of family members in the family realm [4]. Hence, family and healthcare practitioners should scrutinize the intertwined relations from family functioning to their emotional development among family members before germane services and interventions are provided.

Although the findings of the current study have external validity due to all the modeling procedures adjusting for the important sociodemographic covariates of parental and children’s gender, age, and education, certain limitations of the study need to be improved in future research. Firstly, the cross-sectional data of this study makes the causality of the study relations impossible. Future studies should employ a longitudinal design to trace how changes in family functioning may affect the changes in emotional development of family members to portray causal relations. Secondly and plausibly, the current study only constructed and investigated the recursive study relations from family functioning to depression in parents and children through their gratitude, which leaves other theoretically feasible structural relations between family functioning, gratitude, and depression in the family context uncharted [3,23,45,80]. For example, the depressive symptoms of parents may lead to the depressive symptoms of their children, which then can together affect their family functioning and contribute to their gratitude, or the gratitude of parents may lead to the gratitude of their children, which can predict their levels of depression and then together contribute to family functioning. Thereby, future research should compare the validity and predictive power of these theoretically feasible relations of family functioning, gratitude, and depression in the family context. Third, the current study only examined the relation between family functioning, gratitude, and depression among parents and children, in which the way family functioning affects other aspects of emotional development, such as happiness, life satisfaction, and anxiety, is yet unknown. Thereby, future investigation should incorporate different positive and negative emotions in a single model to see how family functioning shapes and sways their development in a longitudinal way. Fourth, the participants of parent–child dyads came from a convenient sample, which restricts their representativeness. Therefore, random sampling is needed in future research to generate representative data and more externally validated study findings. 

## 7. Conclusions

The current study investigated the complex relations of how positive family functioning affected the emotional development of gratitude and depression in parent–child dyads, in which findings of the structural models and serial mediation analyses in the study supported the linked structural relations from positive family functioning to parental and children’s gratitude and depression as well as the bidirectional effects of parental and children’s gratitude and depression. For this, policymakers should think deeply about how to formulate pro-family and health-related policies and social initiatives to promote and enhance interpersonal integration and cohesion as well as the emotional needs of family members in a home context. In addition, family and healthcare practitioners should deliver their services and interventions to support family and emotional health among family members concertedly and also need to work with policymakers on how to design germane policies and deploy social resources effectively that can better serve the different relational and emotional needs of family members in the family context. Furthermore, the study limitations mentioned above are mainly derived from the cross-sectional data, convenient sampling, and research designs of the current study, which restrict the generalizability and causality of the findings. If future research can address the above-mentioned limitations, a clearer picture of the complex processes of family functioning in contributing to the emotional development of its family members can be reached.

## Figures and Tables

**Figure 1 healthcare-13-00147-f001:**
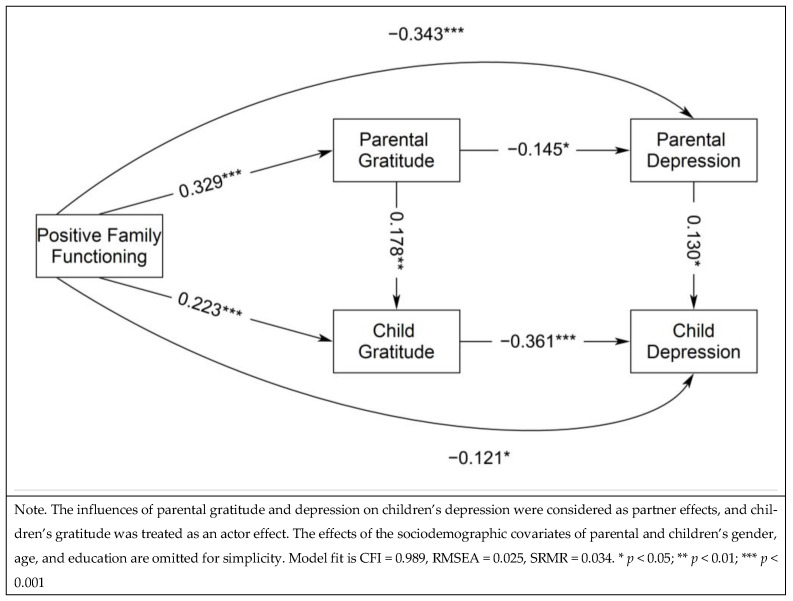
Structural Relations Predicting Children’s Depression from Positive Family Functioning, Parental and Children’s Gratitude, and Parental Depression (Model 1).

**Figure 2 healthcare-13-00147-f002:**
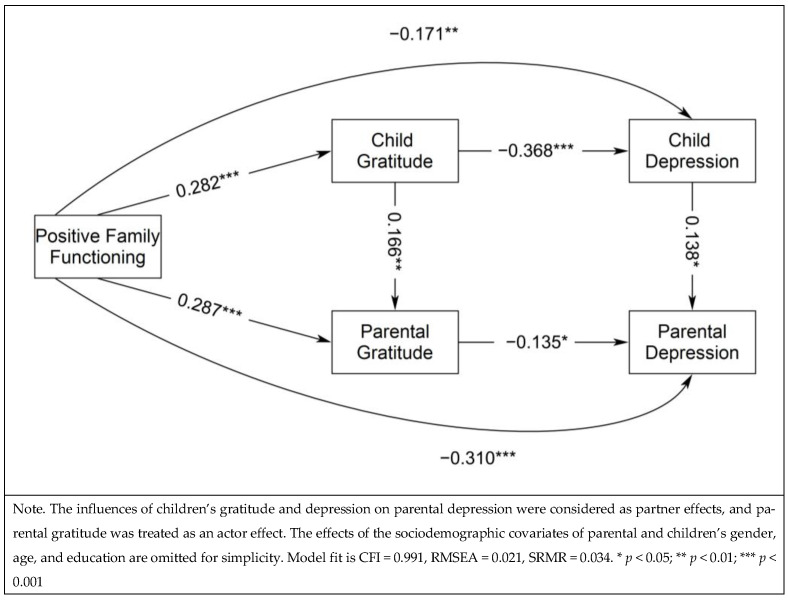
Structural Relations Predicting Parental Depression from Positive Family Functioning, Children’s and Parental Gratitude, and Children’s Depression (Model 2).

**Table 1 healthcare-13-00147-t001:** Sociodemographic Characteristics of The Study Sample of Parent–Child Dyads.

Sociodemographic Variables		Mean/Frequency	SD/Proportion
1.	Parent gender		
	Female	247	0.797
	Male	63	0.203
2.	Parent Age	46.280	7.062
3.	Parent education	2.374	0.956
4.	Child gender		
	Female	182	0.587
	Male	128	0.413
5.	Child age	16.022	1.778
6.	Child education	4.358	1.553

Note. Parent education was measured on a 5-point scale (1 = primary school or below, 2 = secondary school, 3 = associate degree or diploma, 4 = university degree, and 5 = postgraduate degree), and child education was measured on an 8-point scale (1 = primary school or below, 2 = secondary one, 3 = secondary two, 4 = secondary three, 5 = secondary four, 6 = secondary five, 7 = secondary six, and 8 = postsecondary school).

**Table 2 healthcare-13-00147-t002:** Partial Correlations of the Study Variables.

		1	2	3	4	5
1.	Positive Family Functioning	--				
2.	Parental Gratitude	0.298 ***	--			
3.	Child Gratitude	0.256 ***	0.231 ***	--		
4.	Parental Depression	−0.390 ***	−0.250 ***	−0.154 **	--	
5.	Child Depression	−0.244 ***	−0.127 *	−0.382 ***	0.225 ***	--

Note. The gender, age, and education of parents and children were controlled in the partial correlation analysis. * *p* < 0.05; ** *p* < 0.01; *** *p* < 0.001.

**Table 3 healthcare-13-00147-t003:** Indirect Effects Between The Relations of Positive Family Functioning and Parental and Children’s Depression by Serial Mediational Analysis.

Model 1					
	Predictor	Mediator/Mediating Path	Outcome	Indirect Effect	*Z*-value
1.	Positive family functioning	Parental gratitude	Parental Depression	−0.043	−2.403 *
2.	Positive family functioning	Child gratitude	Child depression	−0.079	−3.430 ***
3.	Positive family functioning	Parental depression	Child depression	−0.047	−2.346 *
4.	Positive family functioning	Parental gratitude -> Parental depression	Child depression	−0.006	−1.714 ^+^
5.	Positive family functioning	Parental gratitude -> Child gratitude	Child depression	−0.020	−2.642 **
Model 2					
	Predictor	Mediator/Mediating Path	Outcome	Indirect Effect	Z-value
1.	Positive family functioning	Child gratitude	Child depression	−0.104	−4.336 ***
2.	Positive family functioning	Parental gratitude	Parental depression	−0.039	−2.291 *
3.	Positive family functioning	Child depression	Parental depression	−0.024	−2.056 *
4.	Positive family functioning	Child gratitude -> Child depression	Parental depression	−0.014	−2.247 *
5.	Positive family functioning	Child gratitude -> Parent gratitude	Parental depression	−0.006	−1.825 ^+^

^+^ *p* < 0.1; * *p* < 0.05; ** *p* < 0.01; *** *p* < 0.001.

## Data Availability

The raw data supporting the conclusions of this article will be made available by the authors upon request.

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
