# Peer review of "Intergenerational Transaction of Emotional Health in Collective Family Context: Family Functioning, Parental and Children’s Gratitude, and Their Depression"

_healthcare, 2025, doi:10.3390/healthcare13020147_

Round 1
Reviewer 1 Report
Comments and Suggestions for Authors
Dear Author,
thank you for this topic, I consider it very important as evidence-based research on family studies.
I recommend some completions, through questions or suggestions:
- what programme did you use for analyses? SPSS? PSPP? The abstract and the methodology parts need completion!
- what is the scale of the variables? Must be included in the methodology! Correlation tests can be applied just for ordinal or scales!!!
- if the control variable is gender, with 2 categories, it is the correct method to separate into subgroups and apply the correlation test separately for females and men.
Please, be sure, that the partial correlation test is correct on the scale perspective of the variables!!!
- the text needs some corrections in 44, 254 rows.
Author Response
Dear Reviewer 1,
I have now revised the manuscript titled “Intergenerational Transaction of Emotional Health in the Collective Family Context: Family Functioning, Parental and Children's Gratitude, and Their Depression”, which is submitted to Healthcare. The revision is conducted according to the three reviewers’ comments, for which my responses are below:
#1.1.- what programme did you use for analyses? SPSS? PSPP? The abstract and the methodology parts need completion!
Reply 1.1: As mentioned in section ‘4.3. Analytic Techniques’of the first version of this manuscript, the modeling procedures were fitted in Mplus that have written in the manscritpt: “The modeling procedures were fit in Mplus 8.7 by maximum likelihood estimation with Monte Carlo integration by setting <Integration = Montecarlo> [1]. (p. 7, lines 1120-1121)”
#1.2) - what is the scale of the variables? Must be included in the methodology! Correlation tests can be applied just for ordinal or scales!!!
Reply 1.2: The scales used to measure the study variables of positive family functioning, parental and children’s gratitude, and their depression were fully elaborated in the section of ‘4.2. Measures’ (p.6, lines 918-1106). In addition, correlation analysis can not only be applicable for continuous variables (ordinal or interval/ratio) but also be allowable to examine the association between one continuous variable and one dichotomous variable (dummy) assuming a binary value (e.g., 0 and 1 or 1 and 2), in which this type of correlation analysis is called biserial correlation or point-biserial correlation, depending on the nature of the dichotomous variable. In this study the correlation between parental and children’s gender (1 and 2) with their depression, for example, can be regarded as point biserial correlation, as gender is a naturally dichotomous variable.
# 1.3) - if the control variable is gender, with 2 categories, it is the correct method to separate into subgroups and apply the correlation test separately for females and men.
Reply 1.3: Please see Reply 1.2.
# 1.4) Please, be sure, that the partial correlation test is correct on the scale perspective of the variables!!!
Reply 1.4: Please see Reply 1.2.
#1.5) - the text needs some corrections in 44, 254 rows.
Reply 1.5: Corrected
Regards
The author
Reviewer 2 Report
Comments and Suggestions for Authors
The paper is well structured, but some improvements are suggested below:
Manuscript ID: healthcare-3336935
Title: Intergenerational Transaction of Emotional Health in the Collective Family Context: Family Functioning, Parental and Children's Gratitude, and Their Depression
Global Appreciation: The paper approaches an interesting, original, and actual issue. It is well written, with good English, and organized following the structure of the academic article. The content and methods are robust and dense, and the results are well presented in appropriate tables and diagrams. Some weaknesses were found, which made understanding some components and proposals difficult.
To help improve the manuscript some details will be described following:
Abstract: the aim is long and confusing; methods and tools of data collection were not presented, and then move on to the results. The last sentence should present conclusions and probably recommendations instead of saying that implications will be discussed.
Introduction: The author uses several times assumptions like “I expected” or “I intended”, which, in my opinion, can evidence a tendency that can skew the results. The aims are not clear, being mentioned several times in the text. It is not explicit what is the main objective of the study, and this makes for a complex introduction.
Theoretical Framework of the Study: some concepts need to be defined, more precisely what means home unit. Also, in page 2, line 88, the author needs to verify if the sentence “psychological and psychological performances” is really what is intended to refer.
Methods: the tools for data collection are clearly presented, but the way they were applied is not described. Thus, the recruitment process of the dyads and the way data were collected should be detailed. In page 6, lines 259-260, when referring to the selection of twins, maybe the use of “the first born” can be better than the older since there is not a considerable age difference (only a few minutes, probably). In addition, it is recommended to refer to the number of cases of twins. In line 278, the concept of “multi-informant approach also needs to be clarified.
Results: In Table 1, the Mean doesn´t make sense for the majority of the variables. It is appropriate only for age, as well as the SD. So, this table needs to be replaced because the important is to show the N of individuals in each group generated by the independent variables presented.
Discussion: It is well organized, with limitations well described. Nevertheless, the author needs to reflect on the use of the concepts of negative and positive emotions since emotions are physiological reactions, and they can originate good or bad sensations, but it is different from being positive or negative. The suggestion is to support this classification based on specific literature.
Given the nature of the study and the results obtained, it is recommended to present a conclusion with a good synthesis of the main findings.
The Table 1A appear as appendix, and I don’t understand the option to put it in this part. Why not in the results? Maybe better.
I hope my suggestions can help to improve this paper, which presents a good topic.

Author Response
Dear Reviewer 2,
I have now revised the manuscript titled “Intergenerational Transaction of Emotional Health in the Collective Family Context: Family Functioning, Parental and Children's Gratitude, and Their Depression”, which is submitted to Healthcare. The revision is conducted according to the three reviewers’ comments, for which my responses are below:
#2.1) Global Appreciation: The paper approaches an interesting, original, and actual issue. It is well written, with good English, and organized following the structure of the academic article. The content and methods are robust and dense, and the results are well presented in appropriate tables and diagrams. Some weaknesses were found, which made understanding some components and proposals difficult.
To help improve the manuscript some details will be described following:
Reply 2.1: Thank you.
#2.2) Abstract: the aim is long and confusing; methods and tools of data collection were not presented, and then move on to the results. The last sentence should present conclusions and probably recommendations instead of saying that implications will be discussed.
Reply 2.2: Now the whole Abstract is rewritten, which the sub-headings of ‘Background’, ‘Methods’, ‘Results’, and ‘Conclusions’ are added.
#2.3) Introduction: The author uses several times assumptions like “I expected” or “I intended”, which, in my opinion, can evidence a tendency that can skew the results.
Reply 2.3: Now all the presentations of “I expected” or “I intended” are replaced by “the current study is intended to…” or “this study is planned to…” for precluding personal perceptions.
#2.4) The aims are not clear, being mentioned several times in the text. It is not explicit what is the main objective of the study, and this makes for a complex introduction.
Reply 2.4: The main objectives of the study are presented in the part of ‘1. Introduction’, which are written as:
“Specifically, the current study aims to investigate how positive family functioning aggregately constructed by parents and children, which refers to the manifestation of supportive family relationships, caring interactions, and efficient communication among family members in the family realm, may contribute to both parental and children’s gratitude and depression positively and negatively. Due to the incompatible nature of gratitude and depression in the perspective of Beck et al.’s cognitive depressive bias and the empirical predictive power of gratitude to depression [2-4], this study is also planned to examine whether parental and children’s gratitude would predict their fewer depression, in which both the emotional outcomes are expected to be affected by positive family functioning. Besides, the transactional model of cognition and emotional development posits that family members may mutually and bidirectionally affect their psychological and behavioral performances [5]. For this, the current study is intended to test whether parental gratitude and depression would positively lead to the development of children’s gratitude and depression, and vice versa. Lastly but importantly, based on the structural relations between family functioning, parental and children’s gratitude, and their depression considered above, this study is also planned to test the serial mediations of the actor and partner effects of parental and children’s gratitude and the partner effect of parental and children’s depression on the actor’s depression in parents or children for dissecting the complex relations between the transactional development of parental and children’s gratitude and depression formulated in the family context.(p.2, lines 266-285)”
#2.5) Theoretical Framework of the Study: some concepts need to be defined, more precisely what means home unit. Also, in page 2, line 88, the author needs to verify if the sentence “psychological and psychological performances” is really what is intended to refer.
Reply 2.5: The term of “home unit” is now change to “a family”, “the family context”, or “the family realm” for more understandability. For “psychological and psychological performances”, now this presentation is changed to “psychological and behavioral performances(Lines 276-277 and 293-294).”
#2.6) Methods: the tools for data collection are clearly presented, but the way they were applied is not described. Thus, the recruitment process of the dyads and the way data were collected should be detailed.
Reply 2.6: The scales and items used to measure Positive Family Functioning, Parental and Child Gratitude, and their Depression, as well as their sociodemographic covariates are described in the section of ‘4.2 Measures’, which has:
“4.2. Measures
Positive family functioning was measured by the 26-item Family Functioning Style Scale (FFSS) [6], which has been commonly used to tap into constructive relationships, healthy interactions, efficient communication, and mutual support among family members in a home environment of Chinese societies with good measurement validation [6, 7]. Example items include “We take pride in even the smallest accomplishments of family members” and “Our family sticks together no matter how difficult things get.” For enhancing the objectivity and accuracy of evaluating positive family functioning that was collectively established by parents and children in the family context, a multi-informant approach was adopted to collect data from both the parent and child participants. This measurement approach is designed to collect data points from different related parties who may have different perceptions regarding the same social concept(s) being measured [8]. This approach has been verified to increase external validity and reduce the bias of common method variance [8-10]. In this study, excellent Cronbach alphas were obtained for the parent-child dyads, which were α = .963 and .964 for the parent and child participants. The McDonald’s Omega coefficients were also reflective of excellent measurement reliability for both the parent and child participants, êž· = .961 and .964.
Gratitude of parents and children was measured by the 6-item Gratitude Questionnaire [11], which is a well-validated measure and has been used in Chinese societies to measure grateful thoughts and emotions [12, 13]. The reliability values for the parent and child participants were α = .821, and .837, respectively, which indicate a well adequate level. The McDonald’s Omega coefficients for the parent and child participants were also highly reliable, êž· = .853 and .835.
Depression of parents and children was measured by the 6-item depression subscale of the Brief Symptom Inventory (BSI) [14], which has been commonly used to measure psychological distress and mental disorders among various populations [15-17] and has been well validated in the Chinese context [18]. The Cronbach alpha coefficients obtained in this study were excellent for both the parent and child participants, which were α = .932 and .885, respectively. The McDonald’s Omega reliability coefficients were also excellent for both the parent and child participants, which had êž· = .931 and .893, indicating an excellent level of internal consistency.
Sociodemographic Covariates of parents and children include their gender, age, and educational attainment. Gender of parents and children is a dichotomous variable (1 = male, 2 = female), and the age of parents and children is a count variable. In addition, educational attainment of parents and children is an ordered-categorical variable, in which the educational variable for parents are coded: 1 = primary school or below, 2 = secondary school, 3 = associate degree or diploma, 4 = university degree, and 5 = postgraduate degree; and the educational variable for children is coded: 1 = primary school or below, 2 = secondary one, 3 = secondary two, 4 = secondary three, 5 = secondary four, 6 = secondary five, 7 = secondary six, and 8 = postsecondary school.”
In addition, the sampling and data collection processes are now further elaborated in the section of ‘4.2 Sample’, which has:
“4.1 Sample and Data
The data for analysis in the current study was based on a community sample of 310 Chinese parent-child dyads, who were recruited with the help of the Hong Kong Young Women’s Christian Association (HKYWCA), one of the largest NGOs in Hong Kong. The parent participants were the main caregivers in the family, ensuring their close interactions and influences on the development of their children. The child participants of the participating families were aged between 13 and 21 years old, which means they were in their critical development period, susceptible to the influences of family context. To increase the variance of the child participants, if the participating family had two or more target adolescent children eligible for the study, the one who had just passed her or his birthday was selected. However, if the family had a twin of eligible children suitable for the study, the firstborn was selected to increase randomization. In the sampling procedures, the author first introduced the management of HKYWCA to the study purpose and sampling selection criteria of this research. After that, directors and social workers in respective service units of HKYWCA helped to invite eligible families to take part in the study. These service units cover family services, school social work, and youth service centers in HKYWCA.
During data collection, the parental and child versions of the questionnaires were enclosed in a survey package and sent to the participating families that were eligible for the study through HKYWCA. In the data collection process, social workers in HKYWCA first helped publicize and recruit eligible Chinese parent-youth dyads who were willing to participate in the study. Then the parent-version and child-version questionnaires coded with the same identifiers were sent to the participating families, respectively. After completion of the questionnaires, each participating family was rewarded with a cash purchase coupon worth one hundred Hong Kong dollars for compensating their efforts and time used in participation of the study. After data collection, the management of HKYWCA arranged to send back the completed parent-version and child-version questionnaires to the author of the current study. The study was approved by the ethical review committee of City University of Hong Kong.”
#2.7) In page 6, lines 259-260, when referring to the selection of twins, maybe the use of “the first born” can be better than the older since there is not a considerable age difference (only a few minutes, probably). In addition, it is recommended to refer to the number of cases of twins.
Reply 2.7: Now the term of the older one in twins is changed to “the firstborn”. However, I cannot provide the number of twin cases due to requesting the participating families to ask “the firstborn” to answer the child-version questionnaire is an instruction given to the families for increasing the randomness of sampling, and the participation in the current study is voluntary. Hence, recording the number of twin cases is infeasible in the current study, for which I think this sampling practice is not theoretically impactful on the modeling procedures and the study findings.
#2.8) In line 278, the concept of “multi-informant approach also needs to be clarified.
Reply 2.8: Now “multi-informant approach” to collect data for positive family functioning is further elaborated, which has:
“Positive family functioning was measured by the 26-item Family Functioning Style Scale (FFSS) [6], which has been commonly used to tap into constructive relationships, healthy interactions, efficient communication, and mutual support among family members in a home environment of Chinese societies with good measurement validation [6, 7]. Example items include “We take pride in even the smallest accomplishments of family members” and “Our family sticks together no matter how difficult things get.” For enhancing the objectivity and accuracy of evaluating positive family functioning that was collectively established by parents and children in the family context, a multi-informant approach was adopted to collect data from both the parent and child participants. This measurement approach is designed to collect data points from different related parties who may have different perceptions regarding the same social concept(s) being measured [8]. This approach has been verified to increase external validity and reduce the bias of common method variance [8-10]. In this study, excellent Cronbach alphas were obtained for the parent-child dyads, which were α = .963 and .964 for the parent and child participants. The McDonald’s Omega coefficients were also reflective of excellent measurement reliability for both the parent and child participants, êž· = .961 and .964.(p.6, Lines 1273-1288). ”
#2.9) Results: In Table 1, the Mean doesn´t make sense for the majority of the variables. It is appropriate only for age, as well as the SD. So, this table needs to be replaced because the important is to show the N of individuals in each group generated by the independent variables presented.
Reply 2.9: I think the reviewer misunderstood the presentations of Table 1, in which, as the reviewer said “the important is to show the N of individuals in each group”. Hence, in a descriptive table, we need to present the Mean and SD for continuous variables, e.g., age and educational attainment, and also need to present the proportion and frequency (percentage) for those categorical variables, e.g., gender (1= male, 2=female). Now, I have modified the presentation of Table 1 to make it more easily readable.
#2.10) Discussion: It is well organized, with limitations well described. Nevertheless, the author needs to reflect on the use of the concepts of negative and positive emotions since emotions are physiological reactions, and they can originate good or bad sensations, but it is different from being positive or negative. The suggestion is to support this classification based on specific literature.
Reply 2.10: Now the part of ‘Discussion’ is manifestly revised to elaborate how family functioning contribute to the development of parental and children’s gratitude and depression, in which the positive and negative characteristics and impacts of gratitude and depression are further described, especially for the family context.
#2.11) Given the nature of the study and the results obtained, it is recommended to present a conclusion with a good synthesis of the main findings.
Reply 2.11: Now the part “Conclusion” is added.
#2.12) The Table 1A appear as appendix, and I don’t understand the option to put it in this part. Why not in the results? Maybe better.
Reply: 2.12: As the effects of parental and children’s sociodemographic covariates regarding gender, age, and education are not the study focus of the current research, hence I think it is better to place the table (Table S2) in Appendix for precision and concision. In addition, I have now also placed the Mplus Code (Table S1) in Appendix for references to readers.
# 2.13) I hope my suggestions can help to improve this paper, which presents a good topic.
Reply 2.13: Thank you.
Regards
The author
Reviewer 3 Report
Comments and Suggestions for Authors
The article discusses the results of a study conducted in China on a purposive sample of 310 Chinese parent-child couples. It deals with establishing the relationship between parent-child gratitude and parent-child depression in the context of family functioning and well-being (emotional support and transmission). The author used already proven research tools and examined positive family functioning, parent-child gratitude and parent-child depression, and used gender, age and education as independent variables in his analyses.
According to the author: “This study is the first of its attempts to investigate how family functioning may contribute to parental and children’s gratitude and depression, in which parental and children’s gratitude and depression are believed to mutually affect each other and mediate the relation between positive family functioning and the actor’s depression in parents or children" (lines 422-426).
It turns out that through the analyses presented here, intuitive knowledge (about the mutual positive/negative influence of parents and children) has been confirmed. The results fit in with the assumptions of the transactional model of parent-child relationships in one home. Thus, the author's research adds to the existing knowledge about the mutual influence and emotional relations of family members and their psychological well-being. What's more, they incorporate a broader construct - family funcation - into the analysis.
In my opinion, the article is well prepared. It has a logical structure: introduction to the problem, methodological issues, presentation of the research and discussion of the results. It has a very good literature background. The author correctly presents and discusses the results of his study. Conclusions are consistent with the evidence and arguments presented and relate to the hypotheses established. The tables and graphs are clear and illustrate the results of the study well (with a small exception described below).
My comments:
1. the sociodemographic characteristics othe parents - there is an error in the description of the parents' gender. In Table 1 male =247 and female = 63, while in the description it is the other way around (lines 329-330).
2. I suggest a clearer presentation of the results in Table 1 using four columns N M % SD or two N/M %/SD. In this way there will be a unified record (it is now in the description 79.7% , and in the table .797 and so on)
3. I suggest that the last paragraph regarding the limitations of the study and suggestions for future research (lines 506-528) be separated in a separate paragraph 7. Limitations
4. I would expect more specific guidance on what actions should be taken by the state or health care to “help family members establish positive cognitive processes for their development of gratitude and alleviation of depression” (lines 469-440).
Also for ”healthcare and family practitioners ought to design and implement programming and services at individual, family, and community levels for the enhancement of positive emotions, including gratitude, to tackle the prevalent harms of depression (lines 453-455) - Examples of designing and implementing family policies and interventions?
The article is worth publishing after the suggestions are implemented
Author Response
Dear Reviewer 3,
I have now revised the manuscript titled “Intergenerational Transaction of Emotional Health in the Collective Family Context: Family Functioning, Parental and Children's Gratitude, and Their Depression”, which is submitted to Healthcare. The revision is conducted according to the three reviewers’ comments, for which my responses are below:
#3.1.) 1. the sociodemographic characteristics othe parents - there is an error in the description of the parents' gender. In Table 1 male =247 and female = 63, while in the description it is the other way around (lines 329-330).
Reply 3.1: I have checked the presentation of parental gender, which is now correct.
#3.2) 2. I suggest a clearer presentation of the results in Table 1 using four columns N M % SD or two N/M %/SD. In this way there will be a unified record (it is now in the description 79.7% , and in the table .797 and so on)
Reply 3.2: Now Table 1 is revised to have a clearer presentation, in which Mean/ Frequency and SD/ Proportion are listed to present the continuous (e.g., Age) and categorical variables (e.g., Gender).
#3.3) 3. I suggest that the last paragraph regarding the limitations of the study and suggestions for future research (lines 506-528) be separated in a separate paragraph 7. Limitations
Reply 3.3: Now the limitations of the study are put in the newly-added part of ‘7. Conclusion’.
#3.4) 4. I would expect more specific guidance on what actions should be taken by the state or health care to “help family members establish positive cognitive processes for their development of gratitude and alleviation of depression” (lines 469-440).
Reply3.4 : The part of ‘Discussion’ is now revised to be more organized, instructive, and implicative based on the findings of the current study, in which the presentation to “help family members establish positive cognitive processes for their development of gratitude and alleviation of depression” is now extended to:
“These findings respond to the importance of cultivating a supportive and healthy interpersonal context in the family realm to help family members establish positive cognitive processes for their development of gratitude and alleviation of depression in the processes of implementing healthcare policies and services. This includes letting family members value the importance of cohesive and supportive family relationships, appreciating the proper ways of life experiences and encounters in the family context, and establishing appropriate strategies to handle negative emotions properly.” (p.11, Lines 1772-1779).
#3.5) Also for ”healthcare and family practitioners ought to design and implement programming and services at individual, family, and community levels for the enhancement of positive emotions, including gratitude, to tackle the prevalent harms of depression (lines 453-455) - Examples of designing and implementing family policies and interventions?
Reply 3.5: Now more detailed elaborations for how healthcare and family practitioners can promote family and emotional health in the family realm are mentioned in the part of ‘Discussion.’ Nevertheless, I think to list out examples or steps to instruct healthcare and family practitioners for how to specifically implement an intervention or service or policy to enhance family and emotional health of family members is improper for the nature of an academic paper. This is because a malleable area should be left to healthcare and family practitioners to think about what strategies, approaches, and directions to plan and design their interventions and services based on the findings of the current study and their professional background.
#3.6) The article is worth publishing after the suggestions are implemented
Reply 3.6: Thank you.
Regards
The author
Round 2
Reviewer 2 Report
Comments and Suggestions for Authors
Dear Author,
Congratulations on your efforts. The manuscript improved. Nevertheless, some crucial suggestions need to be incorporated to obtain a manuscript with good scientific quality.
More details are in the attached file, following your responses.
Best regards,

Author Response
Dear Reviewer 2,
I have read the comments of reviewer 2 and have revised the manuscript titled “Intergenerational Transaction of Emotional Health in the Collective Family Context: Family Functioning, Parental and Children's Gratitude, and Their Depression” according to some new comments, but not all, as certain comments, I found, are less justifiable. My responses are below:
~For Reviewer 2:
#2.1) Global Appreciation: The paper approaches an interesting, original, and actual issue. It is well written, with good English, and organized following the structure of the academic article. The content and methods are robust and dense, and the results are well presented in appropriate tables and diagrams. Some weaknesses were found, which made understanding some components and proposals difficult.
To help improve the manuscript some details will be described following:
Reply 2.1: Thank you.
#2.2) Abstract: the aim is long and confusing; methods and tools of data collection were not presented, and then move on to the results. The last sentence should present conclusions and probably recommendations instead of saying that implications will be discussed.
Reply 2.2: Now the whole Abstract is rewritten, which the sub-headings of ‘Background’, ‘Methods’, ‘Results’, and ‘Conclusions’ are added.
Comment: Yes, some changes were made, and it was improved. Nevertheless, the methods continue to be weak. The type of study and tools for data collection were not mentioned. Also, conclusions are longer than results, which is not the better way, as well as results are vague.
Reply 2.2A: For the reviewer’s mentioning the Methods, the selection criteria of Parent-Child Participants, Sampling Procedures, and Data Collection are all elaborated in detail as possible as a research paper submitted to an academic journal can in the Part of ‘4. Methods’ (as at the same time a scholar needs to balance the requirements of concision, precision, and standard of a journal research paper requires). As I have mentioned before, the measurement tools have also been elaborated in Section ‘4.2. Measures’. In fact, I feel confused by the reviewer that he/she has agreed that the current revised manuscript has already clearly incorporated and mentioned the Measures/ Scales/ Tools used in the study in #2.6 by replying that “Yes, the tools / scales and reliability test were well described in the first version” , but he/she replied in this point that I have not done any of requirement. I hope the reviewer can read the revised manuscript carefully and thoughtfully, although I appreciate his/her reviewing efforts in this reviewing process.
Besides, the reviewer said the part of ‘Conclusion’ is longer than the part of ‘Discussion’, which I cannot agree with his/her point, in which the part of ‘Conclusion’ contains “505 words” and the part of ‘Discussion’ contains “1265 words.” For this, I am not sure of his/her claim of the terms ‘longer’ or ‘shorter.’ Nevertheless, I have not tried to remove the contents of “Limitations of the study” to the part of ‘Discussion’ to make a balance.
#2.3) Introduction: The author uses several times assumptions like “I expected” or “I intended”, which, in my opinion, can evidence a tendency that can skew the results.
Reply 2.3: Now all the presentations of “I expected” or “I intended” are replaced by “the current study is intended to…” or “this study is planned to…” for precluding personal perceptions.
#2.4) The aims are not clear, being mentioned several times in the text. It is not explicit what is the main objective of the study, and this makes for a complex introduction.
Reply 2.4: The main objectives of the study are presented in the part of ‘1. Introduction’, which are written as:
“Specifically, the current study aims to investigate how positive family functioning aggregately constructed by parents and children, which refers to the manifestation of supportive family relationships, caring interactions, and efficient communication among family members in the family realm, may contribute to both parental and children’s gratitude and depression positively and negatively. Due to the incompatible nature of gratitude and depression in the perspective of Beck et al.’s cognitive depressive bias and the empirical predictive power of gratitude to depression [2-4], this study is also planned to examine whether parental and children’s gratitude would predict their fewer depression, in which both the emotional outcomes are expected to be affected by positive family functioning. Besides, the transactional model of cognition and emotional development posits that family members may mutually and bidirectionally affect their psychological and behavioral performances [5]. For this, the current study is intended to test whether parental gratitude and depression would positively lead to the development of children’s gratitude and depression, and vice versa. Lastly but importantly, based on the structural relations between family functioning, parental and children’s gratitude, and their depression considered above, this study is also planned to test the serial mediations of the actor and partner effects of parental and children’s gratitude and the partner effect of parental and children’s depression on the actor’s depression in parents or children for dissecting the complex relations between the transactional development of parental and children’s gratitude and depression formulated in the family context.(p.2, lines 266-285)”
Comment: Lines 266-285, neither in the first nor in the second version correspond to it. These lines correspond to the methods section. Nevertheless, in page 2, introduction we can find this. It is suggested aims should be clearer and presented in the final of the introduction and background, before the hypothesis. Without references and more synthetic and objective. In this way, it is confusing.
Reply 2.4A: As mentioned in the previous reply, the revised manuscript has presented the study Purpose and Aims more specifically in the part of ‘1. Introduction’, which has been written:
“As informed by the model of cognition-emotion integration and recent family research findings [8, 11, 12], both cognition and emotional development of family members are first cultivated and formed through the relational and interaction processes and experiences in the family realm and then sway their cognitive and emotional responses to be performed in external social settings. It is expected in this study that family functioning collectively constructed by parents and children may lead to their development of gratitude and depression Specifically, the current study aims to investigate how positive family functioning aggregately constructed by parents and children, which refers to the manifestation of supportive family relationships, caring interactions, and efficient communication among family members in the family realm, may contribute to both parental and children’s gratitude and depression positively and negatively. Due to the incompatible nature of gratitude and depression in the perspective of Beck et al.’s cognitive depressive bias and the empirical predictive power of gratitude to depression [13-15], this study is also planned to examine whether parental and children’s gratitude would predict their fewer depression, in which both the emotional outcomes are expected to be affected by positive family functioning. Besides, the transactional model of cognition and emotional development posits that family members may mutually and bidirectionally affect their psychological and behavioral performances [16]. For this, the current study is intended to test whether parental gratitude and depression would positively lead to the development of children’s gratitude and depression, and vice versa. Lastly but importantly, based on the structural relations between family functioning, parental and children’s gratitude, and their depression considered above, this study is also planned to test the serial mediations of the actor and partner effects of parental and children’s gratitude and the partner effect of parental and children’s depression on the actor’s depression in parents or children for dissecting the complex relations between the transactional development of parental and children’s gratitude and depression formulated in the family context.”
By the way, I do not agree with the reviewer’s suggestion that when introducing and presenting the study purpose and aims, it should be “without references.” In fact, I find that the reviewer would like to use many perplexed and subjective terms in the review comments, such as “vague,” “more synthetic and objective,” and “confused,” which lack concrete evidence to claim his/her suggestions. For this, I cannot totally agree with what the comments claimed by the reviewer, although I admit that many comments are constructive for the revision of the manuscript. Frankly, I hope the reviewer carefully and accurately reads the revised manuscript thoroughly and makes sure he/she has grasped important logics and theoretical arguments, methodology, statistical procedures, and findings of the current study.
#2.5) Theoretical Framework of the Study: some concepts need to be defined, more precisely what means home unit. Also, in page 2, line 88, the author needs to verify if the sentence “psychological and psychological performances” is really what is intended to refer.
Reply 2.5: The term of “home unit” is now change to “a family”, “the family context”, or “the family realm” for more understandability. For “psychological and psychological performances”, now this presentation is changed to “psychological and behavioral performances(Lines 276-277 and 293-294).”
Comment: Ok, home unit was replaced by the indicated concepts. Regarding the psychological performances, it was changed in page 2, now lines 91-92. None of the lines indicated here corresponds to the mentioned as this lines refers to ethical procedures and reliability tests.
Reply 2.5A: All the concepts of “family realm” and “psychological and psychological performances” have been replaced with “a family,” “the family context,” and “psychological and behavioral performances.” Please note that due to the revision, the Line Numbers are also changed.
#2.6) Methods: the tools for data collection are clearly presented, but the way they were applied is not described. Thus, the recruitment process of the dyads and the way data were collected should be detailed.
Reply 2.6: The scales and items used to measure Positive Family Functioning, Parental and Child Gratitude, and their Depression, as well as their sociodemographic covariates are described in the section of ‘4.2 Measures’, which has:
“4.2. Measures
Positive family functioning was measured by the 26-item Family Functioning Style Scale (FFSS) [6], which has been commonly used to tap into constructive relationships, healthy interactions, efficient communication, and mutual support among family members in a home environment of Chinese societies with good measurement validation [6, 7]. Example items include “We take pride in even the smallest accomplishments of family members” and “Our family sticks together no matter how difficult things get.” For enhancing the objectivity and accuracy of evaluating positive family functioning that was collectively established by parents and children in the family context, a multi-informant approach was adopted to collect data from both the parent and child participants. This measurement approach is designed to collect data points from different related parties who may have different perceptions regarding the same social concept(s) being measured [8]. This approach has been verified to increase external validity and reduce the bias of common method variance [8-10]. In this study, excellent Cronbach alphas were obtained for the parent-child dyads, which were α = .963 and .964 for the parent and child participants. The McDonald’s Omega coefficients were also reflective of excellent measurement reliability for both the parent and child participants, êž· = .961 and .964.
Gratitude of parents and children was measured by the 6-item Gratitude Questionnaire [11], which is a well-validated measure and has been used in Chinese societies to measure grateful thoughts and emotions [12, 13]. The reliability values for the parent and child participants were α = .821, and .837, respectively, which indicate a well adequate level. The McDonald’s Omega coefficients for the parent and child participants were also highly reliable, êž· = .853 and .835.
Depression of parents and children was measured by the 6-item depression subscale of the Brief Symptom Inventory (BSI) [14], which has been commonly used to measure psychological distress and mental disorders among various populations [15-17] and has been well validated in the Chinese context [18]. The Cronbach alpha coefficients obtained in this study were excellent for both the parent and child participants, which were α = .932 and .885, respectively. The McDonald’s Omega reliability coefficients were also excellent for both the parent and child participants, which had êž· = .931 and .893, indicating an excellent level of internal consistency.
Sociodemographic Covariates of parents and children include their gender, age, and educational attainment. Gender of parents and children is a dichotomous variable (1 = male, 2 = female), and the age of parents and children is a count variable. In addition, educational attainment of parents and children is an ordered-categorical variable, in which the educational variable for parents are coded: 1 = primary school or below, 2 = secondary school, 3 = associate degree or diploma, 4 = university degree, and 5 = postgraduate degree; and the educational variable for children is coded: 1 = primary school or below, 2 = secondary one, 3 = secondary two, 4 = secondary three, 5 = secondary four, 6 = secondary five, 7 = secondary six, and 8 = postsecondary school.”
In addition, the sampling and data collection processes are now further elaborated in the section of ‘4.2 Sample’, which has:
“4.1 Sample and Data
The data for analysis in the current study was based on a community sample of 310 Chinese parent-child dyads, who were recruited with the help of the Hong Kong Young Women’s Christian Association (HKYWCA), one of the largest NGOs in Hong Kong. The parent participants were the main caregivers in the family, ensuring their close interactions and influences on the development of their children. The child participants of the participating families were aged between 13 and 21 years old, which means they were in their critical development period, susceptible to the influences of family context. To increase the variance of the child participants, if the participating family had two or more target adolescent children eligible for the study, the one who had just passed her or his birthday was selected. However, if the family had a twin of eligible children suitable for the study, the firstborn was selected to increase randomization. In the sampling procedures, the author first introduced the management of HKYWCA to the study purpose and sampling selection criteria of this research. After that, directors and social workers in respective service units of HKYWCA helped to invite eligible families to take part in the study. These service units cover family services, school social work, and youth service centers in HKYWCA.
During data collection, the parental and child versions of the questionnaires were enclosed in a survey package and sent to the participating families that were eligible for the study through HKYWCA. In the data collection process, social workers in HKYWCA first helped publicize and recruit eligible Chinese parent-youth dyads who were willing to participate in the study. Then the parent-version and child-version questionnaires coded with the same identifiers were sent to the participating families, respectively. After completion of the questionnaires, each participating family was rewarded with a cash purchase coupon worth one hundred Hong Kong dollars for compensating their efforts and time used in participation of the study. After data collection, the management of HKYWCA arranged to send back the completed parent-version and child-version questionnaires to the author of the current study. The study was approved by the ethical review committee of City University of Hong Kong.”
Comment: Yes. But only the way the questionnaire was applied and the families are recruited are not clear. It was my previous point and continues to be my doubt. Did you apply questionnaires in paper or online format? How families were selected? It was a probabilistic or a convenience or snowball sample? These are the questions. It would be clarified to improve the quality of the text.
Reply 2.6A: Now the part of “4. Methods” has further elaborated to explain the sampling and data collection procedures as well as the format of the questionnaires used, which has:
“4. Methods
4.1. Sample and Data
The data for analysis in the current study was based on a community sample of 310 Chinese parent-child dyads, who were recruited with the help of the Hong Kong Young Women’s Christian Association (HKYWCA), one of the largest NGOs in Hong Kong. The parent participants were the main caregivers in the family, ensuring their close interactions and influences on the development of their children. The selection criteria of sampling the parent-child participants include the child participants in the participating families should be in the age range between 13 and 21 years old, which means they were in their critical development period, susceptible to the influences of family context; and the parent participants should be the main caregivers to ensure their knowledge of the family context and the development of their children. In addition, if the participating family had two or more target adolescent children eligible for the study, the one who had just passed her or his birthday should be selected, which was to increase the variance of the child participants,. However, if the family had a twin of eligible children suitable for the study, the firstborn should be selected to increase randomization. In the sampling procedures, the purposive sampling approach was adopted to align with the sampling selection criteria mentioned above, in which the author first introduced the management of HKYWCA to the study purpose and sampling selection criteria set for this study. After that, directors and social workers in respective service units of HKYWCA helped to invite eligible families to take part in the study. These service units cover family services, school social work, and youth service centers in HKYWCA.
During data collection, the parental and child versions of the printed questionnaires were enclosed in a survey package and sent to the participating families that were eligible for the study through HKYWCA. In the data collection process, social workers in HKYWCA first helped publicize and recruit eligible Chinese parent-youth dyads who were willing to participate in the study. Then the parent-version and child-version questionnaires coded with the same identifiers were sent to the participating families, respectively. After completion of the questionnaires, each participating family was rewarded with a cash purchase coupon worth one hundred Hong Kong dollars for compensating their efforts and time used in participation of the study. After data collection, the management of HKYWCA arranged to send back the completed parent-version and child-version questionnaires to the author of the current study. The study was approved by the ethical review committee of City University of Hong Kong.”
#2.7) In page 6, lines 259-260, when referring to the selection of twins, maybe the use of “the first born” can be better than the older since there is not a considerable age difference (only a few minutes, probably). In addition, it is recommended to refer to the number of cases of twins.
Reply 2.7: Now the term of the older one in twins is changed to “the firstborn”. However, I cannot provide the number of twin cases due to requesting the participating families to ask “the firstborn” to answer the child-version questionnaire is an instruction given to the families for increasing the randomness of sampling, and the participation in the current study is voluntary. Hence, recording the number of twin cases is infeasible in the current study, for which I think this sampling practice is not theoretically impactful on the modeling procedures and the study findings.
#2.8) In line 278, the concept of “multi-informant approach also needs to be clarified.
Reply 2.8: Now “multi-informant approach” to collect data for positive family functioning is further elaborated, which has:
“Positive family functioning was measured by the 26-item Family Functioning Style Scale (FFSS) [6], which has been commonly used to tap into constructive relationships, healthy interactions, efficient communication, and mutual support among family members in a home environment of Chinese societies with good measurement validation [6, 7]. Example items include “We take pride in even the smallest accomplishments of family members” and “Our family sticks together no matter how difficult things get.” For enhancing the objectivity and accuracy of evaluating positive family functioning that was collectively established by parents and children in the family context, a multi-informant approach was adopted to collect data from both the parent and child participants. This measurement approach is designed to collect data points from different related parties who may have different perceptions regarding the same social concept(s) being measured [8]. This approach has been verified to increase external validity and reduce the bias of common method variance [8-10]. In this study, excellent Cronbach alphas were obtained for the parent-child dyads, which were α = .963 and .964 for the parent and child participants. The McDonald’s Omega coefficients were also reflective of excellent measurement reliability for both the parent and child participants, êž· = .961 and .964.(p.6, Lines 1273-1288). ”
Comment: Ok. But it is the same that already were in the first version and, in this second, it is located in lines 279-294 of the page 6. The problem is that the concept of “multi-informant approach” continues equally unclear. Are it referring to a diverse via of data collection in different formats or referring to researchers involved in participants recruitment. Sorry, but it is not clear and is confusing for the readers.
Reply 2.8A: The concept of the “multi-informant approach” to collect data has been elaborated in the section of “4.2. Measures,” which has:
“4.2. Measures
Positive family functioning was measured by the 26-item Family Functioning Style Scale (FFSS) [8], which has been commonly used to tap into constructive relationships, healthy interactions, efficient communication, and mutual support among family members in a home environment of Chinese societies with good measurement validation [8, 58]. Example items include “We take pride in even the smallest accomplishments of family members” and “Our family sticks together no matter how difficult things get.” For enhancing the objectivity and accuracy of evaluating positive family functioning that was collectively established by parents and children in the family context, a multi-informant approach was adopted to collect data from both the parent and child participants. This measurement approach is designed to collect data points from different related parties who may have different perceptions regarding the same social concept(s) being measured [59], which means both the responses from the parent and child participants were collected and averaged to construct a combined measure of positive family functioning. This approach has been verified to increase external validity and reduce the bias of common method variance [59-61]. In this study, excellent Cronbach alphas were obtained for the parent-child dyads, which were α = .963 and .964 for the parent and child participants. The McDonald’s Omega coefficients were also reflective of excellent measurement reliability for both the parent and child participants, êž· = .961 and .964.”
in which I have highlighted the explanations for the reviewer. In fact, adequate academic readers are knowledgeable of what is a multi-informant approach to collect data.
#2.9) Results: In Table 1, the Mean doesn´t make sense for the majority of the variables. It is appropriate only for age, as well as the SD. So, this table needs to be replaced because the important is to show the N of individuals in each group generated by the independent variables presented.
Reply 2.9: I think the reviewer misunderstood the presentations of Table 1, in which, as the reviewer said “the important is to show the N of individuals in each group”. Hence, in a descriptive table, we need to present the Mean and SD for continuous variables, e.g., age and educational attainment, and also need to present the proportion and frequency (percentage) for those categorical variables, e.g., gender (1= male, 2=female). Now, I have modified the presentation of Table 1 to make it more easily readable.
Comment: Sorry, it is not a misunderstanding. Is the correct sense. You have a number of cases from female sex and male sex, a categorial variable. For example, if you had several groups and were comparing the number of females and males in each group, the mean and the standard deviation, could make sense, not here in this table. It is not correct. Frequency is correct, mean no. And the same for parent education. Maybe you have n or F for each level of education, no mean and SD.
Reply 2.9A: For Table 1, it is a Descriptive Statistics Table for portraying the mean levels and SDs of continuous variables (Age and Education) and Frequency and Proportions of categorical variables (Gender) that are the demographic covariates of the parent and child participants: Gender (categorical), Age (Continuous), and Education (Continuous). I guess the reviewer would like to do group comparison tests, e.g., t-tests, ANOVA, or Chi-Square Comparisons, in which, for example, he/she wants to take gender as the group variable and parental education and age as the outcome variables for the analysis and present the results in this table (Table 1). However, it is non-justifiable in the current study to conduct such group comparison tests, which are not the study purpose of the current study. To my training in advanced statistical procedures and having sought advice from my colleagues of strong quantitative methodology background, I cannot see the incorrect way that the reviewer claimed regarding this descriptive table (Table 1) and I cannot understand why he/she would like to impose such a tendency requiring “comparing the number of females and males in each group, the mean and the standard deviation, could make sense, not here in this table. It is not correct. Frequency is correct, mean no. And the same for parent education. Maybe you have n or F for each level of education, no mean and SD.” In fact, to conduct “Group Comparison Tests,” which are basic inferential statistical modeling procedures, is only justifiable when a theoretical argument is present for the existence of a group difference that needs to be tested, which should be based on necessary theorizing and are generally to be tested before conducting more complicated modeling procedures. However, this is not the study purpose of the current research, and I cannot see the necessity and justification for what the reviewer claimed to complicate Table 1 that is for descriptive purposes. In fact, I disagree with the statistical procedures suggested by the reviewers in this reviewing process, as they commented, and I am sorry to find they even ignored the existence of ‘biserial correlation’ for the association between a dummy and continuous variable.
#2.10) Discussion: It is well organized, with limitations well described. Nevertheless, the author needs to reflect on the use of the concepts of negative and positive emotions since emotions are physiological reactions, and they can originate good or bad sensations, but it is different from being positive or negative. The suggestion is to support this classification based on specific literature.
Reply 2.10: Now the part of ‘Discussion’ is manifestly revised to elaborate how family functioning contribute to the development of parental and children’s gratitude and depression, in which the positive and negative characteristics and impacts of gratitude and depression are further described, especially for the family context.
Comment: The concepts of positive and negative emotions keep equal. The additional sentence refers to other ideas, not clarifying these concepts. But it is a common misconception in studies referring to emotions without a deep knowledge on emotions.
Reply 2.10A: The purpose of the current study is to investigate how family functioning affect the development of gratitude and depression in parents and children, but not a topic on the reviwer so-called focusing on “the concepts of positive and negative emotions”. In fact, the study has tried to elaborated clearly the nature of “gratitude” and “depression”, which include in the part of ‘Discussion’:
“Evidently, the results of the current study found that positive family functioning did significantly predict parental and children’s higher gratitude and their fewer depression. In fact, gratitude and depression are important positive and negative emotions [5, 7, 53], which bear prolonged influences on human physical and psychological health. This is because emotional development and responses are directly related to human neurobiological, behavioral, psychological, and physical representations and health [45, 74, 75]. Thereby, to analyze and clarify how the family context, e.g., family functioning, affects gratitude and depression in parents and their children is of important research contributions to the literature [2]. For gratitude, it is an inherent emotional disposition related to one’s expression of thankfulness and appreciation of what they have and encounter in life [62]. Research has found that gratitude is associated with a number of benefits, such as increased optimism, self-confidence, resilience, happiness, life satisfaction, and physical health [56], and can also preclude potential emotional harms, such as depression, anxiety, psychological distress, and suicidality [56, 76]. The current study found that the collective and constructive family context, that is positive family functioning, is significantly promotive of gratitude among parents and children, which then contributes to their fewer depression. These findings respond to the importance of cultivating a supportive and healthy interpersonal context in the family realm to help family members establish positive cognitive processes for their development of gratitude and alleviation of depression in the processes of implementing healthcare policies and services. This includes letting family members value the importance of cohesive and supportive family relationships, appreciating the proper ways of life experiences and encounters in the family context, and establishing appropriate strategies to handle negative emotions properly.
Moreover, both parental and children’s gratitude are found to predict their fewer depression levels, which concur with existing research results in support of the counter nature of gratitude and depressive symptoms proposed by the cognitive depressive bias perspective [13]. This is because gratitude is a quality of being thankful and showing appreciation for what one has encountered and experienced, which are related to returning kindness to the actor and people and external environments around her or him [6]. Manifestly, gratitude is deemed a type of positive emotions that not only can broaden immediate thinking to a more transcendent and appreciative way for avoiding the happening of depression [15, 53] but also helps employ and utilize personal and social resources flexibly [77], e.g., individual strengths, intellectual and psychological capabilities, and interpersonal relationships. All these can abate the impacts of negative events and failures on incurring depressive symptoms. Besides, pertinent research found that gratitude is positively related to emotional intelligence that is believed to help alleviate depression [78, 79]. For this, gratitude should be promoted in the family realm as a positive emotion to prevent depression. In fact, the characteristics of depressive symptoms include low mood, inability to concentrate, feeling worthlessness, aversion to activity, excessive guilt, and recurrent thoughts of death [7, 37], which may severely impair one’s physical, mental, and interpersonal health and well-being. Therefore, family and healthcare practitioners ought to design and implement programming and services at individual, family, and community levels for the enhancement of positive emotions, including gratitude, to tackle the prevalent harms of depression. This is believed to be an effective strategy before medication and clinical interventions are needed as the therapeutic end for clinical purposes. This is important as the etiology and occurrence of depression result from a complex interaction process related to the social, psychological, and biological spheres [7, 14], in which people who have gone through adverse life events are more susceptible to the development and harms of depression.
Besides, the current study has attempted to investigate whether parental and children’s gratitude and depression may affect each other, which, to the author’s knowledge, is the first research attempt to scrutinize the bidirectional reinforcement of emotional health between parents and children. This research attempt is needed as researchers tended to adopt a unidirectional and parent-driven approach in studying emotional development in the family context [9, 10], which means that the development of gratitude and depression in a family is generally regarded as a transmission from parents to children. Although relevant research has supported this unidirectional and parent-driven approach about the emotional development among parents and children in the family context [14, 25, 30], the transactional model asserts that the emotional and behavioral representations are mutually affected and shaped between parents and children in the family context [16]. Results of the current study support the thesis of the transactional model that parental gratitude and depression contribute to their children’s gratitude and depression, and also children’s gratitude and depression also lead to the development of their parental gratitude and depression. Accordingly, although we conventionally consider parents to be playing the dominating role in shaping the emotional health of their children, results of the current study support the influences of children’s gratitude and depression on their parents’ emotional health. Thus, policymakers, educators, and family and healthcare practitioners should take note of this complex and bidirectionally reinforcing family process regarding the mutuality between parents and children when designing and implementing family policies and interventions. Of specific importance, family and healthcare practitioners should consider interventions and services at both the parental and child levels concomitantly to promote their constructive and healthy interactions and communication as well as emotional management in the family context, which are pivotal to enhance family and emotional health concertedly.”
#2.11) Given the nature of the study and the results obtained, it is recommended to present a conclusion with a good synthesis of the main findings.
Reply 2.11: Now the part “Conclusion” is added.
#2.12) The Table 1A appear as appendix, and I don’t understand the option to put it in this part. Why not in the results? Maybe better.
Reply: 2.12: As the effects of parental and children’s sociodemographic covariates regarding gender, age, and education are not the study focus of the current research, hence I think it is better to place the table (Table S2) in Appendix for precision and concision. In addition, I have now also placed the Mplus Code (Table S1) in Appendix for references to readers.
# 2.13) I hope my suggestions can help to improve this paper, which presents a good topic.
Reply 2.13: Thank you.
Comment: Not all the suggestions were incorporated, as needed, but the paper improved.
Reply 2.13A: I hope the reviewer could carefully read the revised paper and grasp the theoretical rationales, empirical investigations, and systematic arguments of the current study. In fact, the revised manuscript has attempted to incorporate the theoretically and empirically justifiable and important comments as references is improve its academic quality.
Regards
The author
